# Effect of Chia (*Salvia hispanica* L.) Associated with High-Fat Diet on the Intestinal Health of *Wistar* Rats

**DOI:** 10.3390/nu14224924

**Published:** 2022-11-21

**Authors:** Marcella Duarte Villas Mishima, Bárbara Pereira Da Silva, Mariana Juste Contin Gomes, Renata Celi Lopes Toledo, Hilário Cuquetto Mantovani, Vinícius Parzanini Brilhante de São José, Neuza Maria Brunoro Costa, Elad Tako, Hércia Stampini Duarte Martino

**Affiliations:** 1Department of Nutrition and Health, Federal University of Viçosa, Av. Purdue, s/n, Campus Universitário, Viçosa 36570-900, MG, Brazil; 2Department of Microbiology, Federal University of Viçosa, Av. Peter Henry Rolfs, s/n, Campus Universitário, Viçosa 36570-000, MG, Brazil; 3Department of Pharmacy and Nutrition, Federal University of Espírito Santo, Alto Universitário, s/n, Alegre 29500-000, ES, Brazil; 4Department of Food Science, Cornell University, Stocking Hall, Ithaca, NY 14850, USA

**Keywords:** chia seed, western diet, intestinal microbiota, intestinal morphology, SCFA, intestinal functionality

## Abstract

A direct correlation has been reported between excessive fat intake and the development and progression of various enteropathies. Plant foods may contain bioactive compounds and non-digestible dietary fiber, with potential to improve intestinal health. Chia is a good source of dietary fiber and bioactive compounds. Our study evaluated the role of chia flour associated with a high-fat diet (HFD) on colon histomorphometry, intestinal functionality and intestinal microbiome composition and function in *Wistar* rats. The study used 32 young male rats separated into four groups to receive a standard diet (SD) or HFD, with or without chia, for 35 days. At the end of the study, the cecum, cecal content and duodenum were collected. The consumption of chia increased the production of short-chain fatty acids and improved fecal moisture. Chia consumption improved the circular muscle layer in the SD group. The diversity and abundance of intestinal bacteria were not affected, but increased richness was observed in the microbiome of the SD+chia group. Moreover, chia consumption decreased the expression of proteins involved in intestinal functionality. Chia consumption improved intestinal morphology and functionality in young *Wistar* rats but was insufficient to promote significant changes in the intestinal microbiome in a short term of 35 days.

## 1. Introduction

The large consumption of a western diet, which is increased in fat and reduced in vegetables and fiber, is increasingly becoming a big risk factor for several chronic metabolic and inflammatory disorders involving different organs. The chronic high-fat diet (HFD) consumption can induce or aggravate many diseases that affect a wide range of organs besides obesity and metabolic syndrome [1,2]. In addition, a direct correlation has been reported between excessive fat intake and the development and progression of various enteropathies, with conditions that lead to the secretion of pro-inflammatory cytokines, which initiate and maintain an inflammatory process, thus causing alteration in the intestinal microbiota, dysfunction of the epithelial barrier and reduced permeability and integrity of the intestinal mucosal barrier [2,3,4].

By manipulating external factors, the plasticity of the microbiota allows the reshaping of the architecture and biological outputs of intestinal microbes to improve human health. A relation between diet and the intestinal microbiota is observed, which leads to the conclusion that dietary factors are the most potent modulators of microbiota composition and function. Intestinal microorganisms, in turn, influence the absorption, metabolism and storage of nutrients, with effects on host physiology [5].

When it reaches the colon, the fiber diet is anaerobically fermented by the intestinal bacteria. Some nutritionally specialized bacteria in the phyla Firmicutes and Actinobacteria are considered to be important in initiating the degradation of the fiber diet, and the continued breakdown is attributed to certain abundant species within the phylum Bacteroidetes, and then they can produce short-chain fatty acids (SCFAs). SCFAs play a significant role in intestinal homeostasis and affect the body [5,6]. The most abundant SCFAs in the intestine are acetic acid, propionic acid and butyric acid [6,7]. Microbiota-accessible carbohydrates provide an energy source for intestinal bacteria, some of which are nutritionally specialized in degrading these carbohydrates and producing certain SCFA [6]. The consequent production of SCFAs benefits the host by serving as both recovered energy from otherwise inaccessible carbohydrates as well as potent regulatory molecules, with vast physiological effects, including energy homeostasis, lipid and carbohydrate metabolism, and suppression of inflammatory signals [5,6].

Plant foods may contain bioactive compounds and dietary fibers with potential effects on intestinal bacterial populations, gastrointestinal motility improvement, intestinal functionality and morphology, increased mucus production, number/diameter of goblet cells, surface area of villi and crypt depth. Such effects seem to result from the increased motility of the digestive tract, which leads to hyperplasia and/or hypertrophy of muscle cells [7,8,9].

Chia (*Salvia hispanica* L.) is a potentially bioactive food. Its consumption may reduce the risk and attenuate metabolic alterations, due to a series of health benefits, such as antidiabetic effects, antitumor potential, immunostimulant activity, antioxidant protection, cardiovascular and liver protection, remission of inflammation and reduced heart fat content [10,11,12]. Chia contains components with diverse actions, whose benefits are mainly due to their high nutritional value, high concentrations of lipids (32.16 g.100 g^−1^), proteins (18.18 g.100 g^−1^) and total dietary fiber (33.37 g.100 g^−1^), besides insoluble fiber (30.47 g.100 g^−1^), minerals and antioxidant compounds beneficial to health [13]. The intra-amniotic (in ovo) administration of a soluble chia extract improved intestinal morphology, affected the intestinal microbiota and positively regulated the gene expression of proteins associated to mineral metabolism [8]. Furthermore, the relationship between chia seed peptides and the regulation of adipogenesis and inflammation has already been demonstrated [14]. Therefore, the objective of the present study was to evaluate the role of chia associated with a high-fat diet on colon histomorphometry, intestinal functionality and microbiome composition and function of young *Wistar* rats.

## 2. Materials and Methods

### 2.1. Sample Material

The seeds (*Salvia hispanica* L.) used in this study were grown in the state of Rio Grande do Sul (Brazil). The seeds were planted in January 2015, and the harvest was carried out in June 2015. The samples were packed and transported in cardboard boxes. Then, the samples were stored in hermetically sealed plastic bags until use, protected from light and frozen (−18 °C ± 1 °C). The seeds were ground up to a particle size of 850 μm to obtain flour, using a knife mill (Marconi Equipment, Brazil), in three replicates. Then, chia flour was packed in polyethylene aluminum bags and stored in a freezer (−18 °C ± 1 °C) until the chia was added to the diet.

### 2.2. Animals and Diets

Thirty-two newly weaned, 21-day-old male rats (Rattus norvegicus, *Wistar*, albinus variation) from the Central Animal Facility of the Center for Biological Sciences and Health, at the Federal University of Viçosa, Minas Gerais, Brazil, were separated into 4 groups (n = 8) randomized by body weight, using the WinPepi Program version 11.65 [15]. The rats were distributed into individual stainless steel cages in a temperature-controlled environment (22 °C) and automatically controlled light and dark cycles of 12 h. The experimental diets were based either on the standard diet (AIN-93G) [16] or high-fat diet (Research Diets, New Brunswick, NJ, USA), with modifications. The animals received their respective experimental diets and deionized water ad libitum.

The standard diet was composed of 20% protein, 30% fat and 50% carbohydrate. The high-fat diet was prepared with the following proportions: 64% fat, 16% protein and 20% carbohydrate. The amount of chia was determined based on the human consumption of 40 g of chia/day (2 tablespoons). The chia seed was added to diets based on its composition, described by Da Silva (2017) [13], of lipids (32.16 g.100 g^−1^), proteins (18.18 g.100 g^−1^), total dietary fiber (33.37 g.100 g^−1^) and carbohydrates (4.59 g.100 g^−1^). The other ingredients were added in quantities sufficient to provide the planned amounts of proteins, lipids, carbohydrates, fiber and calories (Table 1).

Each group received one of the following experimental diets: standard diet (SD), standard diet + chia flour (SD+chia), high-fat diet (HFD) or high-fat diet + chia flour (HFD+chia). Body weight gain and food consumption were monitored weekly during the experimental period. On the 35th day, after 12 h of fasting, the animals were anesthetized with isoflurane (Isoforine, Cristália^®^), and then euthanized by cardiac puncture. Cecum, cecal content and duodenum were collected, weighed and stored at −80 °C prior to analysis. The colon segment was collected, flushed with phosphate buffered saline solution, fixed in formaline for 24 h and kept in 70% ethanol for histological analysis. The following fat depots was weighed: visceral, retroperitoneal, epididymal and abdominal adipose tissues. The adiposity was calculated as a percentage using the following formula: (visceral + retroperitoneal + epididymal + abdominal adipose tissues)/total body weight × 100 [17].

All the experimental procedures with animals were performed in accordance with Directive 86/609/EEC of November 24, 1986, in compliance with the ethical principles for animal experimentation. The study protocol was approved by the Ethics Committee of the Federal University of Viçosa (protocol 20/2017; date of approval: 13 July 2017).

### 2.3. Fecal Moisture

The moisture content in the feces, collected at the end of the experiment, was determined by the gravimetric method. The samples were oven-dried at 105 °C for 24 h [18].

### 2.4. Cecal Content pH

About 1 g of cecum feces was homogenized in 10 mL of distilled water, with the aid of vortex glass spheres. Subsequently, the glass electrode of the pHmeter (Bel, Italy) was inserted. The measurements were performed in duplicate [19].

### 2.5. IgA Quantification

In order to determine sIgA, the standard procedures were followed to prepare the cecal content homogenates, 1:5 (*w*/*v*) . The cecal content samples were defrosted on ice. To prepare the suspensions, 200 mg feces was added to 800 µL of phos-phate-buffered saline and homogenized with the aid of a vortex. It was used an Immunochron enzyme-linked immunosorbent assay (ELISA) to measured the mucosal immunity, based on the cecal content sIgA concentration. [20].

### 2.6. Short Chain Fatty Acids (SCFA) Content

The analysis to determine the content of SCFA followed the methodology proposed by Siegfried and Ruckemmann (1984) [21], with modifications. Briefly, 500 mg of cecum feces were homogenized in MiliQ water, following a vortex shaking protocol. Next, the samples were centrifuged at 12,000× *g* for 10 min. The supernatants were transferred to new tubes, received the addition of calcium hydroxide and cupric sulfate and were vigorously shaken. The samples were frozen and then thawed at room temperature for centrifugation. The supernatant was added with concentrated sulfuric acid and then frozen. Then, the samples were thawed, refrozen and finally thawed for centrifugation at 12,000× *g*, for 10 min. The supernatant was transferred to vials for analysis by high-performance liquid chromatography (HPLC).

The SCFAs were determined in a Dionex Ultimate 3000 Dual detector HPLC ap-paratus (Dionex Corporation, Sunnyvale, CA, USA) equipped with a refractive index detector Shodex RI-101, using a Bio-Rad HPX-87H column, 300 mm × 4.6 mm, main-tained at 45 °C. The conditions that the analyses were performed were as follow: mobile phase sulfuric acid 5 mmol L^−1^, flow rate 0.7 mL min^−1^ and injection volume 20 μL. Acetic, propionic and butyric acids were used to prepare the stock solutions of the standards. The SCFAs were prepared with a final concentration of 10 mmol/L.

### 2.7. Colon Histomorphometry

Semi-serialized histological fragments from proximal colon (3 μm thickness) were obtained on a semi-automated rotating microtome (Leica, Brazil) and stained by the hematoxylin/eosin technique. To measure the length and thickness of the crypts and the thickness of the circular and longitudinal muscle layers twenty random fields per animal were selected. The slides were examined under a AX70 photomicroscope (Olympus, Japan). The ImagePro-Plus^®^ software sys-tem, version 4.5 (Media Cybernetics, Rockville, MD, USA) was used to processed the images.

### 2.8. DNA Extraction and Sequencing

The total genomic DNA was extracted from the cecum content samples, following a mechanical disruption and phenol/chloroform extraction protocol [22]. PCR amplicon libraries targeting the hypervariable V4 region of the 16S rRNA gene were produced using the primers 515F (5′GTGYCAGCMGCCGCGGTAA3′) and 806R (GGACTACNVGGGTWTCTAAT3′) and a barcoded primer set adapted for the Illumina MiSeq platform (Illumina, San Diego, California, CA, USA) [23,24]. Illumina MiSeq was used to load the samples onto an Illumina flow cell for paired-end sequencing reactions in the Environmental Sample Preparation and Sequencing Facility (ESPSF), at the Argonne National Laboratory (Lemont, IL, USA).

The sequences obtained for all samples in the present study were submitted to the Sequence Read Archive (SRA) of the National Center for Biotechnology Information (NCBI), under the accession number PRJNA805271.

The Mothur software system v.1.44.3 was used to data processing and analysis [25]. The R1 and R2 paired-end reads were joined, and sequences smaller than 150 or greater than 300 bp were removed. Sequences with homopolymers with at least 8 nucleotides or containing ambiguous base pairs were also eliminated. Chimera sequences were detected and removed using UCHIME [26]. After cleaning the sequences, they were aligned with the 16S rRNA gene, using the SILVA database v.138 [27].

The Operational Taxonomic Units (OTUs) were grouped according to the sequence similarity cutoff. The coverage of all samples was assessed by Good’s coverage estimator (bacteria > 97%). A normalized data table was used for calculating alpha and beta diversity and the relative abundance of OTUs. Alpha diversity was estimated by using Chao1, Shannon and Simpson indices. Beta diversity between dietary groups was assessed by Principal Coordinate Analysis (PoA), based on the Jaccard dissimilarity index [28]. Metagenome functional predictive analysis was carried out using PICRUSt2 software [29].

### 2.9. Determination of Gene Expression of Proteins Involved in Intestinal Health by Quantitative Reverse Transcriptase Polymerase Chain Reaction (RT-qPCR)

mRNA expression levels of genes in the intestinal tissue (duodenum) which are involved in intestinal health were analyzed by RT-qPCR. The SYBR Green PCR master mix from Applied Biosystems (Foster City, CA, USA) was employed, and the analyses were performed on the StepOne™ Real-Time PCR System (Thermo Fisher Scientific) by means of the measurement system involving SYBR-Green Fluorescence and Primer Express software (Applied Biosystems, Foster City, CA, USA). Sense and antisense primer sequences were ordered (Choma Biotechnologies) to amplify aminopeptidase (AP) (ID: 301368687), sucrose isomaltase (SI) (ID: 301368688), peptide transporter 1 (PepT1) (ID: 301368693) and sodium–glucose transport protein 1 (SGLT1) (ID: 301368686). The relative expression levels of mRNA were normalized to the endogenous control (beta-actin; Table 2). All the steps were performed under open conditions with RNase.

### 2.10. Statistical Analysis

Food consumption, body weight, colonic histomorphometry characteristics and concentrations of SCFA data were first submitted to a Kolmogorov–Smirnov normality test. Next, a one-way analysis of variance (ANOVA) was applied, followed by the Newman–Keuls post hoc test to compare all test groups. The experimental treatments were arranged in a completely randomized design, with eight repetitions. The data are presented as means ± standard deviation, and statistical significance was established at *p* < 0.05. Correlations between biological markers and intestinal parameters were assessed by Pearson’s correlation test. The biochemical parameters and stress oxidative markers were previously carried out and published [30].

Regarding the microbiome results, the Chao1, Shannon and Simpson indexes were used to estimates differences between alpha diversity, ANOVA was used to analyse the differences between the groups. The differences between beta diversity were analyzed by the pairwise PERMANOVA test. Statistically significant *p*-values associated with microbial clades and functions were corrected for multiple comparisons using the Benjamini–Hochberg false discovery rate (FDR) correction. The statistical analysis was performed using SPSS version 20.0. The level of significance was established at *p* < 0.05.

## 3. Results

### 3.1. Data of Animals

The food consumption was higher in the rats fed the standard diet (SD and SD+chia) than in the rats fed HFD (HFD and HFD+chia). The animals that were fed with the HFD containing chia (HFD+chia) presented higher food consumption than those of the HFD group. The animals fed the SD and HFD without chia (SD and HFD) presented lower final weight than those in the groups fed chia. Adiposity, cecum weight and cecal content pH did not differ between the experimental groups. The consumption of chia (SD+chia × SD and HFD+chia × HFD) increased the moisture concentration in the feces of the animals and did not alter the IgA concentration, while the SD group presented the lowest IgA concentration (Table 3). Furthermore, chia consumption increased the concentration of acetic, propionic and butyric acids in the cecal content (Figure 1).

### 3.2. Colonic Histomorphometry Characteristics

The HFD consumption reduced the longitudinal muscle layer, circular muscle layer, crypt length and crypt thickness. On the other hand, when associated with high-fat diet, chia consumption reduced the longitudinal muscle layer. Chia consumption with a standard diet increased the circular muscle layer compared to all the other groups. However, in the high-fat groups, chia consumption reduced this measurement. The crypt length was increased by chia consumption in the group that received HFD, but in the group fed the standard diet, the crypt length and crypt thickness were reduced by chia consumption (Table 4).

### 3.3. Intestinal Microbiota Analysis

Regarding the intestinal microbiota analysis, the sequencing of the 16S rRNA gene from the fecal samples generated 731.297 raw sequences. Following the filtering and cleaning of the sequences, 569.406 sequences of suitable quality were obtained. The Good’s coverage obtained in the samples was >99%, which indicates adequate sequencing coverage. The summary of the sample sequencing data is shown in the Appendix A.

The richness and diversity indexes were used to evaluate the alpha diversity of the microbial samples. No difference was observed in the Chao1, Shannon and Simpson indexes among the experimental groups (Figure 2A–C). However, when assessed pairwise (SD × SD+chia and HFD × HFD+chia), the Chao1 index show an increase in richness in the intestinal microbiota of the animals fed SD+chia compared to the SD group (Figure 2A).

According to the beta diversity analyses, calculated by using Jaccard distances, there was some variation in bacterial communities in response to the consumption of the four types of diet at the level of OTU, phyla and genera, as indicated by the PCoA analysis and PERMANOVA (Figure 3A–C).

The spatial ordination of the OTUs (Figure 3A) indicated differences in the clustering of the samples of the different experimental groups. The data based on their collections of sequences presented differences in the distance metrics among the SD group compared to SD+chia, and between the HFD group and HFD+chia. However, after FDR correction, the clustering of OTU sequences presented no difference. Spatial ordination at the phylum level (Figure 3B) indicated no changes between all treatment groups or between the HFD and HFD+chia groups. Although PERMANOVA identified a significant difference between the clustering of phyla from SD and SD+chia groups, no difference was verified after FDR correction.

The statistical data revealed that before FDR correction, the Lachnospiraceae family was enriched in the group fed HFD+chia compared to the HFD group. In addition, the Muribaculaceae family and the genus *Roseburia* were enriched in the group fed SD+chia compared to the SD group, which indicates a beneficial effect of chia consumption on the SCFA-producing bacteria (Appendix A).

The spatial ordination at the genus level (Figure 3C) presented differences in the distance metrics among the experimental groups. However, as previously observed at the OTU and phylum levels, after FDR correction, these differences were lost. It is important to highlight that the *Corynebacterium* genera concentration differs between the groups fed with HFD and HFD+chia.

The taxonomic analysis of the bacterial community in response to chia flour consumption revealed the existence of 18 phyla, 30 classes, 65 orders, 93 families and 193 genera. The stratification of phyla, genus and the Firmicutes to Bacteroidetes ratio is plotted in Figure 4. Although no difference was found in the relative abundance of the identified phyla and genus between the experimental groups, after FDR correction, we observed the dominancy of phyla Firmicutes (63–68% of relative abundance) and Bacteroidetes (19–22% of relative abundance) composing the intestinal microbiota of young rats (Figure 4A), and the dominancy of genera from the Muribaculaceae family, and *Bacteroides* (Figure 4C), identified after 35 days of treatment. Furthermore, no difference was observed in the Firmicutes to Bacteroidetes ratio between the experimental groups (Figure 4B).

We investigated whether chia treatments affected the genetic capacity of the microbiota and explored possible functional changes. PICRSt2 was used for the functional predictive analysis of the metagenome and revealed the most abundant pathways (Figure 5), but no difference was detected between the treatments.

### 3.4. BBM Functional Proteins

Regarding intestinal functionality, we observed that the group that consumed HFD presented higher gene expression for all intestinal genes evaluated. Furthermore, chia consumption (SD+chia and HFD+chia) reduced PepT1, AP, SGLT1 and SI mRNA gene expression in relation to the control diets (SD and HFD) (Figure 6).

### 3.5. Pearson Correlation Analysis

We performed the Pearson correlation analysis to assess the relationship between changes in intestinal microbial abundance, gut health parameters, oxidative stress markers and biochemical variables. When correlations were assessed, *Bacteroides* were negatively correlated with the Muribaculaceae family (r = −0.487), *Roseburia* (r = −0.480) and butyric acid (r = −0.496). The Lachnospiraceae family was negatively correlated with total cholesterol (TC) (r = −0.507). The Muribaculaceae family was positively correlated with IgA (r = 0.482). *Roseburia* was positively correlated with acetic acid (r = 0.602), propionic acid (r = 0.471), catalase (CAT) (r = 0.485) and malondialdehyde (MDA) (r = 0.495). Acetic acid was positively correlated with propionic acid (r = 0.836), butyric acid (r = 0.649), fecal moisture (r = 0.662) and total antioxidant capacity of plasma (TAC) (r = 0.475), and negatively correlated with TC (r = −0.426). Propionic acid was positively correlated with butyric acid (r = 0.664) and fecal moisture (r = 0.711), CAT (r = 0.590), and negatively correlated with TC (r = −0.515). Butyric acid was positively correlated with fecal moisture (r = 0.581) and TAC (r = 0.486), and negatively correlated with TC (r = −0.423). The fecal moisture was positively correlated with the cecal content pH (r = 0.785) and CAT (r = 0.576), and negatively correlated with TC (r = −0.598). Superoxide dismutase (SOD) was positively correlated with MDA (r = 0.597). Finally, CAT was positively correlated with MDA (r = 0.515) and glucose (r = 0.391), and negatively correlated with TC (r = −0.427) (Figure 7).

## 4. Discussion

Chia is a good source of bioactive compounds and dietary fiber [13,31]. However, the potential effects of chia associated with an inflammatory condition on the intestinal microbiota composition, intestinal morphology and intestinal functionality have not been investigated. Thus, the present study focused on evaluating the effect of chia consumption associated with HFD, for 35 days, on the gut health of young *Wistar* rats. Our study revealed that the intake of chia with the SD or HFD increased the production of acetic acid, propionic acid and butyric acid and increased fecal moisture. The consumption of HFD affects the intestinal morphology, reducing the longitudinal and circular muscle layer and the length and thickness of the crypts. Chia consumption increased the crypt length in the group that received HFD and improved circular muscle layer in the group that received SD. The diversity and abundance of intestinal bacteria were not affected, but increased richness was observed in the intestinal microbiome of animals fed SD+chia compared to the SD group. Additionally, chia consumption reduced the expression of proteins involved in intestinal functionality.

The lower food consumption observed in animals fed HFD can be attributed to higher energy density and greater satiety during the experimental period, as previously reported [12,30]. When chia was consumed with HFD, the food consumption was increased, as was the final weight. In the group that was fed SD, although the food consumption was not altered by chia consumption, the SD+chia group presented higher final weight, but no difference in adiposity was observed between groups. This leads us to assume that there was an increase in the muscle mass of these animals. The study of Grancieri et al. (2022) [32] observed that the incorporation of the digested protein of chia into a normal diet increased body weight and contributed to muscle mass gain in the animals, which impacted weight gain without increasing body fat. The skeletal muscle plays a major role in fatty acid uptake and oxidation, while HFD can increase the susceptibility to loss of muscle mass and the degradation of proteins [33]. Chia consumption, when associated with SD and HFD, increased fecal moisture, probably due to the higher content of dietary fibers present in food. The gel-forming dietary fibers present sufficient water-holding capacity and increase fecal bulking action [34]. The increased weight of the feces makes it easier for them to pass through the colon and be expelled from the body, thus alleviating problems such as constipation. Here, the increased moisture in groups that were fed chia can be attributed to chia’s ability to absorb water, which adds bulk to the feces. Chia secrets a mucilage when it becomes wet, a gelatinous and viscous substance after water absorption, which can increase fecal moisture [34].

In general, a high amount of soluble and insoluble dietary fiber increases intestinal motility and fecal volume, which tends to increase the thickness of muscle layers. The circular muscle layer was higher in the group fed SD+chia than in the SD group. This increase was also observed in female ovariectomized *Wistar* rats [35]. This result is probably due to the increased motility of the digestive tract, enabled by the intact form of the dietary fiber found in chia, besides the formation of gel, promoted by the soluble dietary fiber fraction, which leads to the hypertrophy of muscle cells [36].

The consumption of dietary fiber provides a substrate for microbial activity and affects the intestinal microbiota by altering bacterial fermentation and fermentation products, such as SCFA [5,6]. In our study, increased SCFA production was observed in groups that were fed chia. These changes in total SCFAs indicate that the dietary fiber from chia seed could be utilized by the microbiota [37]. SCFAs, such as butyric acid, acetic acid and propionic acid, are the most abundant fecal metabolites. It is important to mention that acetic acid was the most abundant SCFA, which is in agreement with previous reports by Tamargo et al. (2018) [37]. Different SCFA production patterns are possibly dependent on the type of fiber. These fermentation products are extremely important for host health, since butyric acid, apart from serving as the primary energy source for colonocytes, also improves the integrity of intestinal epithelial cells by promoting tight junctions and cell proliferation. Both acetic acid and propionic acid also aid in anti-inflammatory processes and cytokine production [38]. It is suggested that acetic acid plays an important regulatory role in body weight control and insulin sensitivity, acting as a direct mediator in lipid metabolism and glucose homeostasis [39]. The production of SCFAs can reduce the intestinal pH, which may increase mineral solubility and absorption [8,40]. However, in the present study, the cecal content pH was not altered. In our study, the Pearson’s correlation indicated that SCFAs were positively correlated with fecal moisture and antioxidant capacity, and negatively correlated with cholesterol.

In our study, the consumption of chia did not affect the immunoglobulin A (IgA) concentration. IgA is abundant in the intestine and plays an essential role in the defense of the intestinal mucosa against harmful pathogens. It is suggested that diet and intestinal microbiota are involved in the regulation of IgA production [41,42]. Our result is in agreement with another study that also evaluated the impact of chia consumption on intestinal health and found that there was no change in the concentration of IgA [35]. Nakajima et al. (2020) [43] revealed that the microbiota in the large intestine is involved in IgA induction by dietary fiber, and the amount of IgA was considerably higher in the intestinal contents of mice that were fed a diet with soluble dietary fiber than in those that were fed a diet with insoluble dietary fiber. The concentration of dietary fiber in chia is mainly formed by insoluble fiber [13], which can explain why the IgA concentration was not altered in our animals.

The diversity and abundance of intestinal bacteria were not affected when the Chao1, Shannon and Simpson indexes were calculated among the four experimental groups. However, the Chao1 index indicated increased richness in the intestinal microbiome of the animals fed a standard diet+chia (SD+chia) compared to the SD group, in agreement with our findings from another study [35]. Richness is related to the number of different species, and high richness of the intestinal microbiome is associated with healthy host metabolism, while their absence is aligned with unhealthy outcomes [38,44,45,46]. In the same group (SD+chia x SD), the Muribaculaceae family and the genus *Roseburia* were enriched before FDR correction. Studies suggest that the Muribaculaceae family have members with a functional potential in the intestine, which is the ability to degrade dietary carbohydrates and ferment polysaccharides into SCFAs [47,48]. *Roseburia* was positively correlated with acetic acid and propionic acid and has also been reported to produce SCFAs, mainly propionic acid [38], which is in accordance with our results. Therefore, our results indicated a beneficial effect of chia consumption on the SCFA-producing bacteria. It is important to highlight that *Corynebacterium* genus concentration varies between the groups fed with HFD and HFD+chia. Chia consumption was able to reduce the concentration of the genus. This is a positive event, since *Corynebacterium* can produce phospholipase D, an exotoxin that degrades lipids in cell membranes, which may increase cellular permeability and thereby facilitate the spread of the pathogen in the tissues [49]. In addition, according to the functional analysis of the microbiota, we observed the KEGG metabolic pathways that were more abundant, although no differences were detected between the treatments. This fact can be explained by the short time of chia consumption by the animals (35 days).

When we evaluated the intestinal functionality, we found that the HDF increased the expression of all genes evaluated, and chia consumption (HFD+chia) decreased the gene expression, which was similar to the group that consumed SD+chia. Chia consumption reduced the expression of SI and SGLT1, which are genes related to carbohydrate digestion and absorption. Studies have reported that fasting plasma glucose concentration is reduced after chia consumption [36,50,51]. Thus, chia intake may reduce carbohydrate absorption/digestion through the downregulation of SI and SGLT1 gene expression and the consequent reduction in SI, which suppresses the rapid production of glucose on the surface of the brush border membrane, in combination with decreased glucose absorption by reducing the relative expression of SGLT1 [52]. The same result was observed for AP and PepT1 protein gene expression. The function of PepT1 is to transport peptides to the enterocyte. AP is an exopeptidase responsible for cleaving amino acids from the N-terminus of peptides [53]. In our study, we observed that chia consumption reduced PepT1 and AP gene expression, probably due to the fact that the organism is in homeostasis, which does not require the increase in the expression of these genes to perform its function with regard to protein metabolism.

## 5. Conclusions

Chia consumption in standard and high-fat diet increased the production of short-chain fatty acids (acetic acid, propionic acid and butyric acid) and improved fecal moisture. Furthermore, chia consumption improved intestinal morphology, increasing the circular muscle layer in the SD group and the crypt length in the group that received HFD. Besides this, chia decreased the gene expression of SGLT1, SI, AP and PepT1 in all groups. The 35-day intervention in young male *Wistar* rats did not affect the diversity and abundance of intestinal bacteria but promoted an increase in richness in the SD+chia group. Therefore, further studies using a longer intervention period are needed to clarify the effects of chia on the intestinal microbiome.

## Figures and Tables

**Figure 1 nutrients-14-04924-f001:**
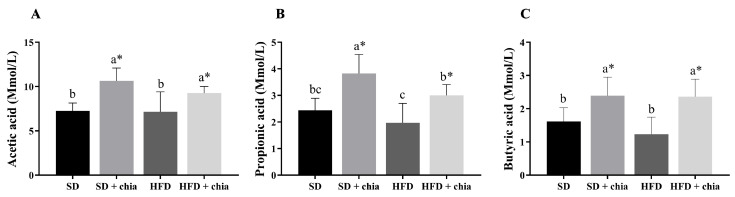
Short-chain fatty acid (SCFA) concentration in cecal contents of young *Wistar* rats, after 35 days of treatment (n = 8). (**A**) Concentration of acetic acid; (**B**) concentration of propionic acid; (**C**) concentration of butyric acid. SD: standard diet; SD+chia: standard diet + chia; HFD: high-fat diet; HFD+chia: high-fat diet + chia. The data were analyzed by one-way ANOVA. Means followed by different small letters in the same row differ significantly, according to Newman–Keuls post hoc test, at 5% threshold of probability. * Indicates differences between the groups by the *t*-test, at 5% probability, in the comparison of the groups that received the same diet, either with or without chia (SD × SD+chia and HFD × HFD+chia).

**Figure 2 nutrients-14-04924-f002:**
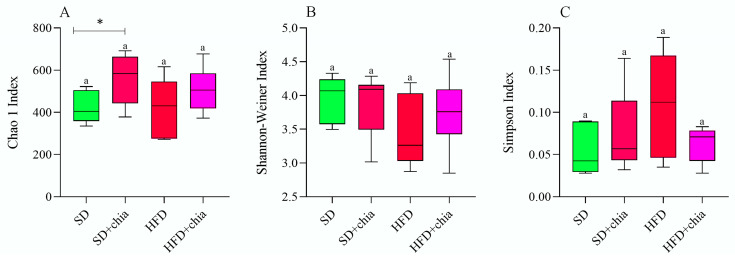
Alpha diversity metrics of bacterial communities in the cecum faces. n = 6 per group. (**A**) Chao1 index; (**B**) Shannon-Weiner Index; (**C**) Simpson Index; SD: standard diet; SD+chia: standard diet+chia; HFD: high-fat diet; HFD+chia: high-fat diet+chia. Treatment groups indicated by the same letter are not significantly different (*p* < 0.05). * Indicates differences between the groups by the *t*-test, at 5% probability, in the comparison of the groups that received the same diet, either with or without chia (SD × SD+chia and HFD × HFD+chia).

**Figure 3 nutrients-14-04924-f003:**
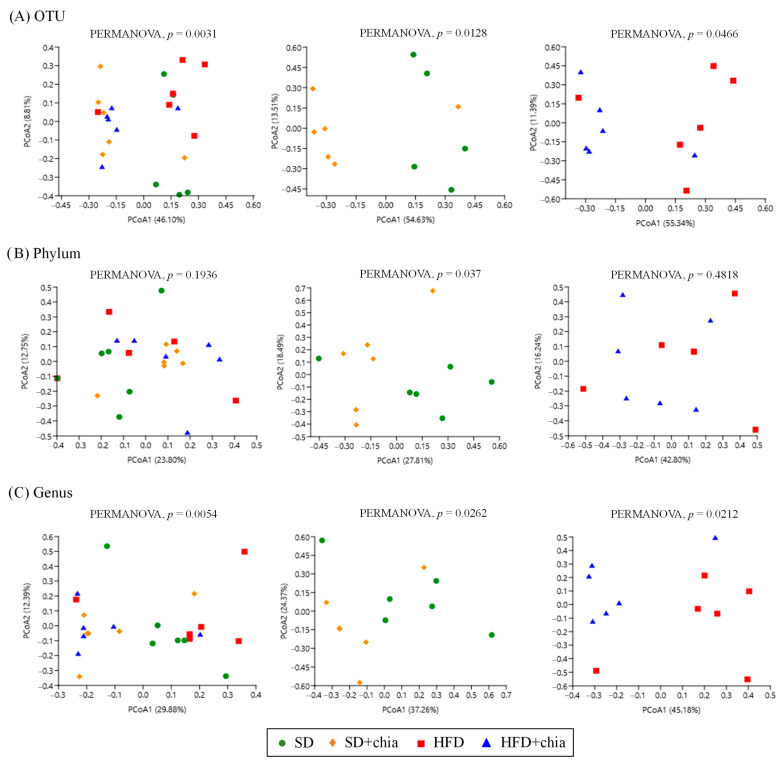
Changes in β-diversity of the cecal microbiome of young *Wistar* rats after 35 days of treatment. (**A**) Principal Coordinate Analysis (PCoA) based on the Jaccard distance at OTU level. (**B**) Principal Coordinate Analysis (PCoA) based on Jaccard distance at phylum level. (**C**) Principal Coordinate Analysis (PCoA) based on Jaccard distance at genus level.

**Figure 4 nutrients-14-04924-f004:**
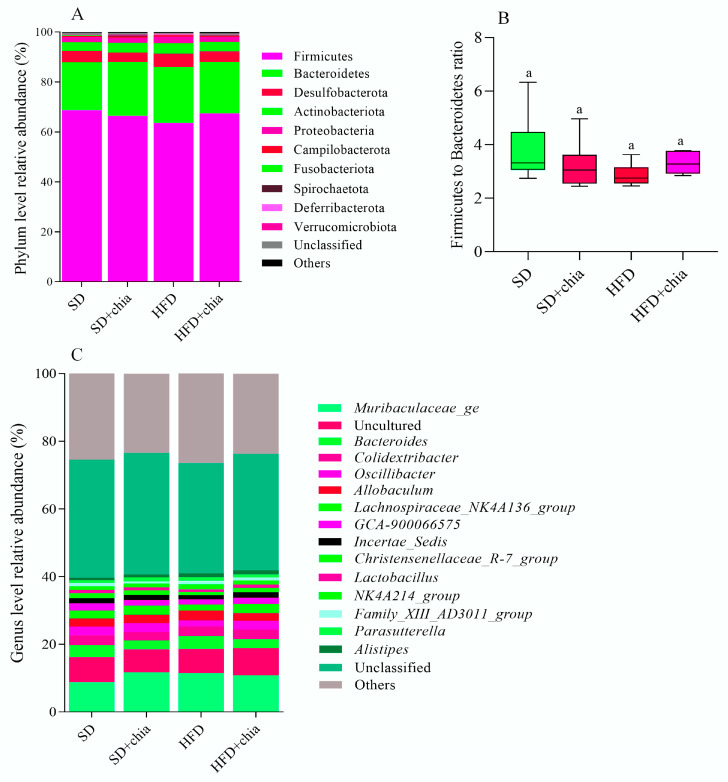
Relative abundances of bacterial microbiota composition at phylum and genera level of young *Wistar* rats, after 35 days of treatment. (**A**) Relative abundance of each identified phylum. (**B**) Firmicutes to Bacteroidetes ratio. (**C**) Genera relative abundance. n = 6/group. SD: standard diet; SD+chia: standard diet + chia; HFD: high-fat diet; HFD+chia: high-fat diet + chia. Only phyla with abundance >0.2% and genera with abundance >1% in at least one group are displayed. The data were analyzed with FDR correction. Treatment groups indicated by the same letter are not significantly different (*p* < 0.05). * Indicates differences between the groups by the *t*-test, at 5% probability, in the comparison of the groups that received the same diet, either with or without chia (SD × SD+chia and HFD × HFD+chia).

**Figure 5 nutrients-14-04924-f005:**
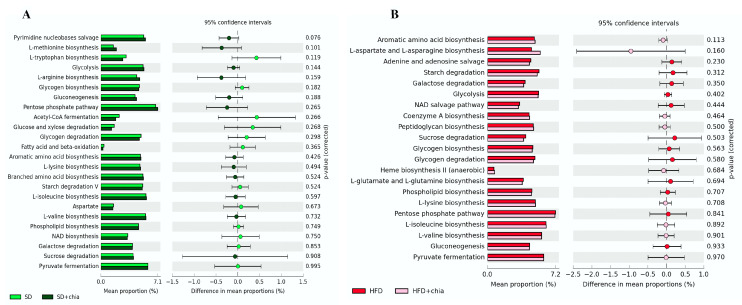
Difference in the relative abundance of the most abundant KEGG microbial metabolic pathways in the microbiota of young *Wistar* rats, after 35 days of treatment. (**A**) Enriched pathways between the SD and SD+chia treatment groups. (**B**) Enriched pathways between the HFD and HFD+chia treatment groups. Extended error bar plot was performed by bioinformatic software (STAMP). Welch’s two-sided test was used and Welch’s inverted was 0.95. n = 6/group. SD: standard diet; SD+chia: standard diet + chia; HFD: high-fat diet; HFD+chia: high-fat diet + chia.

**Figure 6 nutrients-14-04924-f006:**
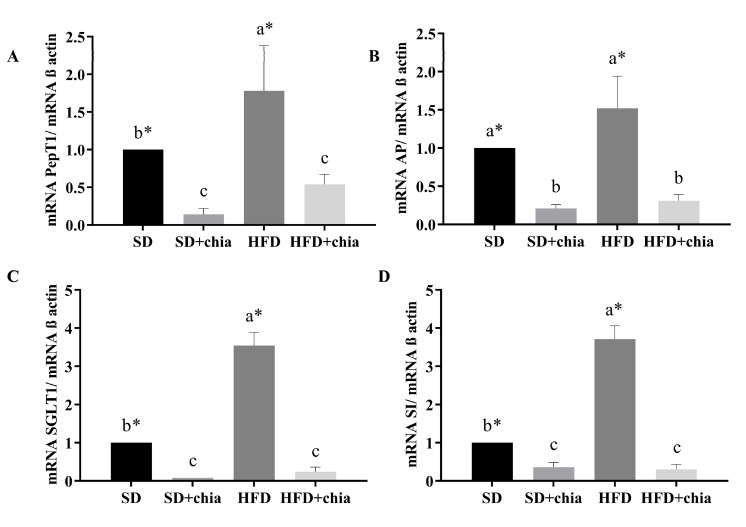
Gene expression of brush border membrane functional proteins in the intestinal tissue. RT-qPCR analysis. (**A**) PepT1 expression, (**B**) AP expression, (**C**) SGLT1 expression and (**D**) SI expression. ST: standard diet; ST+chia: standard diet + chia; HFD: high-fat diet; HF+chia: high-fat diet + chia; PepT1: peptide transporter 1; AP: aminopeptidase; SGLT1: sodium–glucose transport protein 1; SI: sucrose isomaltase. The data were analyzed by one-way ANOVA. Means followed by different small letters in the same row differ significantly, according to the Newman–Keuls post hoc test, at the 5% threshold of probability. * Indicates differences between the groups by the *t*-test, at 5% probability, in the comparison of the groups that received the same diet, either with or without chia (SD × SD+chia and HFD × HFD+chia).

**Figure 7 nutrients-14-04924-f007:**
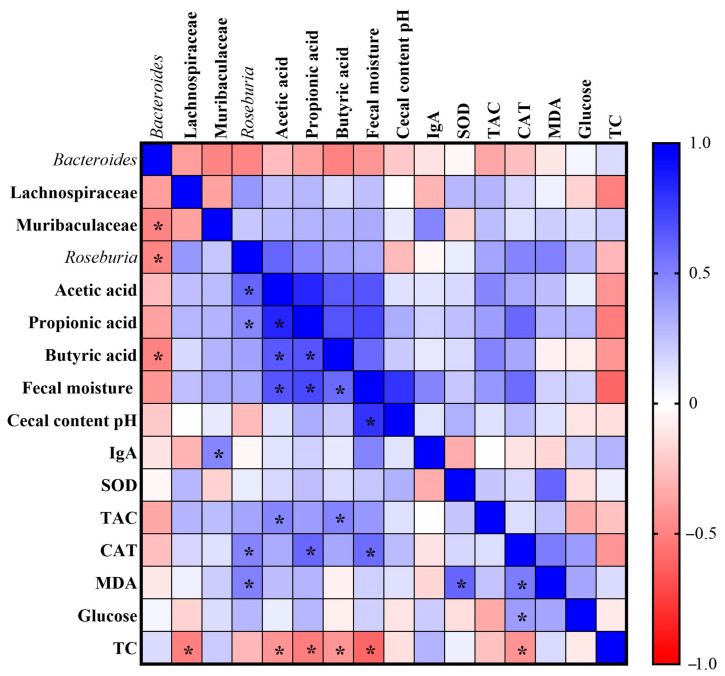
Heatmap of Pearson correlation analysis. IgA: immunoglobulin; SOD: superoxide dismutase; TAC: total antioxidant capacity of plasma; CAT: catalase; MDA: malondialdehyde; TC: total cholesterol. * Indicates statistically significant difference.

**Table 1 nutrients-14-04924-t001:** Compositions of the experimental diets.

	Experimental Diets			
Ingredients (g/kg of Diet)	SD	SD+chia	HFD	HFD+chia
Albumin *	217.90	117.60	217.92	117.60
Chia flour	0.00	416.80	0.00	416.80
Dextrinized starch	132.00	132.00	132.00	115.10
Sucrose	100.00	100.00	100.00	100.00
Lard	0.00	0.00	200.00	200.00
Soybean oil (mL)	134.20	0.00	134.20	0.00
Microcrystalline cellulose	139.20	0.00	139.20	0.00
Mineral mix	35.00	35.00	35.00	35.00
Vitamin mix	10.00	10.00	10.00	10.00
L-cystine	3.00	3.00	3.00	3.00
Choline bitartrate	2.50	2.50	2.50	2.50
Corn starch	221.20	183.10	21.20	0.00
Total calories (kcal)	3700.42 ^b^	3624.75 ^b^	4700.42 ^a^	4624.11 ^a^
Caloric density (kcal g^−1^)	3.70 ^b^	3.62 ^b^	4.70 ^a^	4.62 ^a^

* Considering that albumin is 78% pure. SD: standard diet, HFD: high-fat diet. Means followed by different small letters in the same row differ significantly, according to Newman–Keuls post hoc test, at the 5% threshold of probability.

**Table 2 nutrients-14-04924-t002:** Sequencing primers used in the RT-qPCR analysis.

Genes	Oligonucleotide (5′-3′)	
	Forward	Reverse
Beta-actin	TTCGTGCCGGTCCACACCC	GCTTTGCACATGCCGGAGCC
AP	CTCTCTCCTCAAACACATGAA	AGTTCAGGGCCTTCTCATATTC
SI	CCTCCAGAACACAATCCCTATAC	GGAGAGGTGAGATGGATTAGA
PEPT1	CCTGGTCGTCTCATCATATT	TTCTTCTCATCCTCATCGAACTG
SGLT1	CATCCAGTCCATCACATTAC	CAATCAGGAAGCCGAGAATCA

AP: aminopeptidase; SGLT1: sodium–glucose transport protein 1; SI: sucrose isomaltase; PepT1: peptide transporter 1.

**Table 3 nutrients-14-04924-t003:** Biometric data, consumption data, fecal moisture, cecal content pH and IgA concentration.

	SD	SD+chia	HFD	HFD+chia
Food consumption (g/day)	14.74 ± 0.86 ^a^	15.46 ± 1.20 ^a^	10.86 ± 0.49 ^c^	12.29 ± 0.93 ^b^*
Final weight (g)	221.69 ± 15.25 ^b^	241.10 ± 15.65 ^a^*	208.76 ± 11.95 ^b^	240.11 ± 16.89 ^a^*
Adiposity (% of body weight)	2.32 ± 0.60 ^a^	2.66 ± 0.39 ^a^	2.69 ± 0.49 ^a^	2.96 ± 0.46 ^a^
Cecum weight (g)	4.20 ± 0.71 ^a^	4.76 ± 0.60 ^a^	4.25 ± 0.40 ^a^	4.06 ± 0.83 ^a^
Cecal content pH	7.23 ± 1.28 ^a^	7.33 ± 0.27 ^a^	6.70 ± 0.21 ^a^	7.13 ± 0.45 ^a^
Fecal moisture (%)	16.36 ± 3.86 ^c^	37.65 ± 5.37 ^a^*	14.36 ± 6.41 ^c^	28.01 ± 5.97 ^b^*
IgA (ng)	271.09 ± 95.70 ^b^	555.50 ± 193.67 ^a^	751.83 ± 278.88 ^a^	681.17 ± 155.91 ^a^

Values referring to means ± SD, n = 8/group. SD: standard diet; SD+chia: standard diet + chia; HFD: high-fat diet; HFD+chia: high-fat diet + chia. The data were analyzed by one-way ANOVA. Means followed by different small letters in the same row differ significantly, according to Newman–Keuls post hoc test, at the 5% threshold of probability. * Indicates differences between the groups by the *t*-test, at 5% probability, in the comparison of the groups that received the same diet, either with or without chia (SD × SD+chia and HFD × HFD+chia).

**Table 4 nutrients-14-04924-t004:** Colonic histomorphometry characteristics of young *Wistar* rats after 35 days of treatment.

	SD	SD+chia	HFD	HFD+chia
LML (μm)	49.16 ± 12.48 ^a^	52.65 ± 26.90 ^a^	26.02 ± 11.36 ^b^*	22.68 ± 10.94 ^b^
CML (μm)	111.88 ± 35.84 ^b^	127.44 ± 70.86 ^a^	62.32 ± 24.37 ^c^*	50.01 ± 24.13 ^d^
Crypt length (μm)	180.53 ± 41.33 ^a^*	163.67 ± 44.52 ^b^	103.39 ± 13.97 ^d^	121.51 ± 30.96 ^c^*
Crypt thickness (μm)	45.57 ± 7.64 ^a^*	36.00 ± 13.53 ^b^	26.72 ± 5.67 ^c^	25.41 ± 6.08 ^c^

Values referring to means ± SD, n = 8/group. SD: standard diet; SD+chia: standard diet + chia; HFD: high-fat diet; HFD+chia: high-fat diet + chia; LML: longitudinal muscle layer; CML: circular muscle layer. The data were analyzed by one-way ANOVA. Means followed by different small letters in the same row differ significantly, according to the Newman–Keuls post hoc test, at the 5% threshold of probability. * Indicates differences between the groups by the *t*-test, at 5% probability, in the comparison of the groups that received the same diet, either with or without chia (SD × SD+chia and HFD × HFD+chia).

## Data Availability

Not applicable.

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
