# Peer review of "Effect of Chia (Salvia hispanica L.) Associated with High-Fat Diet on the Intestinal Health of Wistar Rats"

_nutrients, 2022, doi:10.3390/nu14224924_

Round 1

Reviewer 1 Report

The paper has an interesting and relevant subject, is well structured, and is well written. 

Line 18: This is misleading because the authors used the chia flower (I realized reading further).  

Line 97: please specify the provider for WinPepi Program version 11.65.

Line 407-408: please discuss this statement. The same information is in line 456. It is not worth repeating it! Why is the protein expression reduced?

Line 411-412: please discuss this statement. Why was the mass of the rats fed with chia higher?

Overall the discussion section should be improved. 

Also, I recommend inserting a graphical abstract!

Please be consistent, add a space before "g" (grams) all over the manuscript, and use italics for the name of the microbes. 

Please provide a more detailed conclusion. 

Reviewer 2 Report

Review for nutrients-2036154

The manuscript by Marcella Duarte Villas Mishima et al., entitled “Effect of chia (Salvia hispanica L.) associated with high-fat diet on the intestinal health of Wistar rats” the research article reported the findings that Chia consumption improved intestinal morphology and functionality in young Wistar rats but not to promote significant changes in intestinal microbiome in a short term of 35 days. The authors reported that the consumption of chia increased the production of short-chain fatty acids and improved fecal moisture. Chia consumption improved the circular muscle layer. The diversity and abundance of intestinal bacteria were not affected, but the increased richness was observed in the gut microbiome of the SD+chia group. Besides, chia consumption decreased the expression of proteins involved in intestinal functionality. The manuscript is well-written, and the cited references are appropriate. However, some minor remarks should be taken by authors under consideration.

Comments:

1.     Author should mention “Figure 1” in the Figure legend of Fig.1.

2.     In Figure 1, the “Y” axis legend is missing. The authors should correct this.

3.     In Figures 1 & 6, for all the graphs authors should mention the group’s name in “X” axis legend. It will be a better serve to the readers.

Author Response

Response to Reviewer 2 Comments

Point 1: Author should mention “Figure 1” in the Figure legend of Fig.1.

We appreciate the consideration. We mentioned “Figure 1” in the Figure legend of Fig.1. Please, see page 7, line 254.

Point 2: In Figure 1, the “Y” axis legend is missing. The authors should correct this.

We appreciate the consideration. We thank for suggestion and agree with the reviewer. We added “Y” axis legend. Please, see page 7, line 253.

Point 3: In Figures 1 & 6, for all the graphs authors should mention the group’s name in “X” axis legend. It will be a better serve to the readers.

We appreciate the consideration. We thank for suggestion and agree with the reviewer. We changed the legends and added the group’s name in “X” axis. Please, see page 7, line 253 and page 12, line 355.

Round 2

Reviewer 1 Report

According to the comments, I consider the author's responses suitable and the paper's changes. So I endorse publication.